# An Efficient Electrochemical Biosensor to Determine 1,5-Anhydroglucitol with Persimmon-Tannin-Reduced Graphene Oxide-PtPd Nanocomposites

**DOI:** 10.3390/ma16072786

**Published:** 2023-03-30

**Authors:** Guiyin Li, Zhide Zhou, Zhongmin Wang, Shiwei Chen, Jintao Liang, Xiaoqing Yao, Liuxun Li

**Affiliations:** 1College of Chemistry, Guangdong University of Petrochemical Technology, Guandu Road, Maoming 525000, China; 2School of Life and Environmental Sciences, Guangxi Key Laboratory of Information Materials, Guilin University of Electronic Technology, Guilin 541004, China; 3Solid Tumour Target Discovery Laboratory, Translational and Clinical Research Institute, Newcastle University Centre for Cancer, Faculty of Medical Sciences, Newcastle University, Newcastle upon Tyne NE2 4HH, UK

**Keywords:** 1,5-anhydroglucitol, electrochemical biosensor, graphene-based nanomaterials, persimmon tannin, pyranose oxidase

## Abstract

1,5-Anhydroglucitol (1,5-AG) is a sensitive biomarker for real-time detection of diabetes mellitus. In this study, an electrochemical biosensor to specifically detect 1,5-AG levels based on persimmon-tannin-reduced graphene oxide-PtPd nanocomposites (PT-rGO-PtPd NCs), which were modified onto the surface of a screen-printed carbon electrode (SPCE), was designed. The PT-rGO-PtPd NCs were prepared by using PT as the film-forming material and ascorbic acid as the reducing agent. Scanning electron microscopy (SEM), transmission electron microscopy (TEM), ultraviolet–visible spectroscopy (UV–vis), and X-ray diffraction (XRD) spectroscopy analysis were used to characterise the newly synthesised materials. PT-rGO-PtPd NCs present a synergistic effect not only to increase the active surface area to bio-capture more targets, but also to exhibit electrocatalytic efficiency to catalyze the decomposition of hydrogen peroxide (H_2_O_2_). A sensitive layer is formed by pyranose oxidase (PROD) attached to the surface of PT-rGO-PtPd NC/SPCE. In the presence of 1,5-AG, PROD catalyzes the oxidization of 1,5-AG to generate 1,5-anhydrofuctose (1,5-AF) and H_2_O_2_ which can be decomposed into H_2_O under the synergistic catalysis of PT-rGO-PtPd NCs. The redox reaction between PT and its oxidative product (quinones, PTox) can be enhanced simultaneously by PT-rGO-PtPd NCs, and the current signal was recorded by the differential pulse voltammetry (DPV) method. Under optimal conditions, our biosensor shows a wide range (0.1–2.0 mg/mL) for 1,5-AG detection with a detection limit of 30 μg/mL (S/N = 3). Moreover, our electrochemical biosensor exhibits acceptable applicability with recoveries from 99.80 to 106.80%. In summary, our study provides an electrochemical method for the determination of 1,5-AG with simple procedures, lower costs, good reproducibility, and acceptable stability.

## 1. Introduction

Diabetes mellitus (DM) is a chronic metabolic disease that often causes severe complications such as coronary heart disease, diabetic neuropathy, and diabetic nephropathy [1,2]. Among the non-genetic biomarkers of DM, 1,5-anhydroglucitol (1,5-AG), one 1-deoxy form of glucose, has been demonstrated to be a useful biomarker for postprandial hyperglycaemia and short-term glucose control [3]. The serum 1,5-AG not only reflects daily glycaemic excursion but also represents postprandial glucose levels in a dynamic manner [4,5]. The normal concentration of 1,5-AG is approximately 12–40 μg/mL. However, serum 1,5-AG concentrations have been found to decrease in patients with DM [6,7,8]. Several methods have been explored to detect the level of serum 1,5-AG in a sensitive and accurate way. Routine techniques, such as the enzymatic method, chromatography, colourimetric assay, and mass spectrometry, have been used in clinical practice [4,9,10,11,12,13]. Currently, the Glycomark^TM^ assay, approved by the United States Federal Drug Administration in 2003, is regarded as the gold standard to determine serum 1,5-AG levels [11,14].

Recently, electrochemical biosensors have become the most promising alternative method for biological sensing due to their cost effectiveness, portability, rapid response, high sensitivity, and selectivity [15,16,17,18]. Moreover, nanomaterials have also attracted more attention in the field of sensing partly owing to their ability to improve the sensitivity and specificity of target detection. Specifically, nanomaterials not only provide a large surface area and a favourable microenvironment but also exhibit remarkable conductivity and biocompatibility [19,20,21,22]. For example, graphene-based nanomaterials, which possess extraordinary mechanical strength, a large specific surface area, and high conductivity, are already being used in several electrochemical applications [23,24,25]. Additionally, metal nanoparticles, such as Au nanoparticles, Ag nanoparticles, Pt nanoparticles, and Pd nanoparticles, have been widely used in electrochemical sensors to increase electron transfer rates and amplify electrochemical signals [26,27,28,29].

Persimmon tannin (PT), which is extracted from green persimmon, is a kind of unique flavonoid polyphenol and contains multiple active groups (ortho-phenolic hydroxyl groups) which allow PT to be involved in antioxidation, protein binding, and the adsorption of metal ions such as Cr (VI), Ag (I), Au (III), Pt (Ⅳ), and Pd (II) [30,31,32,33,34]. Moreover, the process of reduced graphene oxide (rGO) agglomeration can be prevented by π–π and electrostatic interactions between PT and rGO surfaces [35,36]. In our previous study, one non-enzymatic H_2_O_2_ sensor based on a reduction graphene oxide–persimmon tannin–platinum nanocomposite (rGO-PT-Pt) was constructed. The synthesised rGO-PT-Pt non-enzymatic biosensor has been shown to exhibit satisfied electro-catalytic activity toward H_2_O_2_ reduction with a fast amperometric response time (3 s) and a low detection limit (0.26 μM) for H_2_O_2_ detection [37].

In this study, we prepared persimmon tannin–reduced graphene oxide-PtPd nanocomposites (PT-rGO-PtPd NCs) to construct an electrochemical biosensor that specifically detects serum 1,5-AG. By exploiting the large specific surface area of rGO, the adsorption property, and the electro-activation of PT, the catalytic activity of PtPd nanoparticles, and the PT-rGO-PtPd NCs were introduced to catalyze H_2_O_2_ via a synergistic effect. Then, the as-prepared homogeneous PT-rGO-PtPd NCs suspension was modified onto the surface of screen-printed carbon electrodes (SPCEs). Next, pyranose oxidase (PROD), which specifically recognises 1,5-AG, was loaded onto the modified electrodes to construct a 1,5-AG electrochemical biosensor. To assess the electrochemical properties of this 1,5-AG biosensor, we used electrochemical impedance spectroscopy (EIS), cyclic voltammetry (CV), and the differential pulse voltammetry (DPV) method. The experimental conditions were also optimised. Lastly, our 1,5-AG electrochemical biosensor was used to determine the serum 1,5-AG.

## 2. Materials and Methods

### 2.1. Chemicals and Reagents

Graphene oxide (GO, 98%) was obtained from XFNANO Materials Technology Co., Ltd. (Nanjing, China). Chloroplatinic acid (HPtCl_4_, 96%) and palladium nitrate (Pd (NO_3_)_2_, >99%) were purchased from Sinopharm Chemical Reagent Co., Ltd. (Shanghai, China). PT was obtained from Guangxi Gongcheng Huikun Agricultural Products Co., Ltd. (Guilin, China). 1,5-AG PROD was purchased from Sigma-Aldrich (St. Louis, MO, USA). Glucose (99%), ascorbic acid (AA, 99%), uric acid (UA, 99%), and cholesterol (99%) were supplied by Xilong Chemical Co. (Shanghai, China). Ultrapure water (≥18.25 MΩ·cm) was prepared by using a Milli-Q purification system (Milli-Pore, Bedford, MA, USA).

### 2.2. Apparatus

PT-rGO-PtPd NCs and the modified electrodes were characterised by scanning electron microscopy (SEM, Quanta 200, FEI COMPANY, Valley City, ND, USA), transmission electron microscopy (TEM, JEM-2100F, JEOL, Tokyo, Japan), Raman microscopy (RM, Thermo DXRxi, Renishaw, North East Derbyshire, UK), and ultraviolet–visible spectroscopy (UV–vis, UH5300, HITACHI, Tokyo, Japan). A D/Max-2500 diffractometer (Philips, Minato, Japan) was used to collect the X-ray diffraction (XRD) data.

All electrochemical measurements were performed at room temperature on a CHI660D electrochemical workstation (Shanghai Chenhua Instrument, Shanghai, China) with the three typical SPCEs (Nanjing Yunyou Biotechnology Co., Ltd., Nanjing, China) system. One carbon electrode was used as the working electrode (Φ = 3 mm, the electrode area is 0.07 cm^2^), another carbon electrode was used as the auxiliary electrode, and an Ag/AgCl electrode was used as the reference electrode.

CV was performed with a scanning range from −1.0 to 1.0 V and a scanning rate of 100 mV/s in phosphate-buffered saline (PBS) solution (0.1 mol/L Na_2_HPO_4_/KH_2_PO_4_ and 0.1 mol/L NaCl; pH 7.0) containing 5.0 mmol/L K_3_Fe (CN)_6_/K_4_Fe (CN)_6_ and 0.1 mol/L KCl. EIS was performed at 0.24 V from 1 to 100 KHz and at a scan rate of 50 mV/s in PBS solution containing 5.0 mmol/L K_3_Fe (CN)_6_/K_4_Fe (CN)_6_ and 0.1 mol/L KCl. DPV was performed with a potential range of −0.8 to 0.2 V, a scan rate of 100 mV/s, and an amplitude of 50 mV in PBS solution.

### 2.3. Preparation of the PT-rGO-PtPd NCs

The PT-rGO-PtPd NCs were prepared by using PT as the film-forming material and ascorbic acid (AA) as the reducing agent. Firstly, 5.0 mg of single-layer GO was added to 50.0 mL of distilled water to make 0.1 mg/mL GO dispersion mixed through ultrasonic destruction for 120 min. Then, AA was added to the GO dispersion at a ratio of 1:1 before the dispersion was agitated on a constant-temperature magnetic heating agitator for 12 h. Next, the solution was centrifuged to remove the supernatant. The residues were later dried to obtain the rGO powder. Secondly, 20 mg of PT was added into 10 mL of rGO solution (0.1 mg/mL) to be sonicated for 30 min and homogenised to obtain the PT-rGO suspension. Four mL of HPtCl_4_ (0.01 mg/mL) and four mL of Pd (NO_3_)_2_ (0.01 mg/mL) were then added to the above solution. Thirdly, 10 mg of AA was added before the solution was vigorously agitated for 20 h. After that, the mixed solution was centrifuged for 15 min at 10,000 rpm. The black precipitate (i.e., PT-rGO-PtPd NCs) was obtained by removing the supernatant and washed with ultrapure water three times. Lastly, the PT-rGO-PtPd NCs were dried and stored in a refrigerator at 4 °C.

### 2.4. Construction of the Electrochemical Biosensor Based on PT-rGO-PtPd NCs

First, SPCE was placed in 5 mL of H_2_SO_4_ solution (0.5 mol/L) for 10 cycles of CV scanning. Then, the electrode was electrodeposited in 5 mL of 0.01% HAuCl_4_ solution for 120 s [16]. After the Au NP/SPCE was washed with distilled water and then dried, 9.0 μL of PT-rGO-PtPd NCs suspension (0.1 mg/mL) was added dropwise to the surface of Au NP/SPCE and placed in an incubator at 25 °C for 30 min to obtain the PT-rGO-PtPd NCs /Au NP/SPCE. Lastly, 3.0 μL of PROD (0.5 mg/mL) was added dropwise to the surface of the PT-rGO-PtPd NCs /Au NP/SPCE, and the electrode was then air-dried for 3 h to construct the electrochemical biosensor (i.e., PROD/PT-rGO-PtPd NCs/Au NP/SPCE).

### 2.5. Detection of 1,5-AG with the Electrochemical Biosensor

Three μL of 1,5-AG with different concentrations (0.1, 0.3, 0.5, 1.0, 1.5, and 2.0 mg/mL) was added dropwise onto the surface of the PROD/PT-rGO-PtPd NCs/Au NP/SPCE. Then, the mixture was placed in an incubator at 37 °C for 40 min and washed with PBS to obtain the working electrode (i.e., 1,5-AG/PROD/PT-rGO-PtPd NCs/Au NP/SPCE). Later, the working electrode was inserted into 5.0 mL of PBS solution (0.1 mol/L, pH 7.0) and the electrochemical responses were next recorded by using the DPV method with a peak amplitude of 50 mV, a pulse period of 0.5 s, and a voltage range of −0.8–0.2 V.

Additionally, glucose, ascorbic acid, uric acid, and cholesterol were used as the interfering substances. They were respectively added dropwise to each 1,5-AG working electrode. Based on the optimal conditions, electrochemical detection was next performed to investigate the anti-interference ability of the 1,5-AG electrochemical biosensor.

### 2.6. Application in Serum Samples

To validate the capability of the proposed method for analyzing human serum samples, recovery tests were performed via the standard additional method by using the 1,5-AG biosensor. Human serum samples were collected from the Guangxi Key Laboratory of Metabolic Diseases Research in the 924th Hospital of the Chinese People’s Liberation Army (Guilin, China). First, the samples were diluted with 0.1 mol/L PBS ten times to avoid the matrix effect. Approximately 1.5 μL of quantitative 1,5-AG solution (0.50 mg/mL, 1.0 mg/mL, and 2.0 mg/mL) and 1.5 μL of diluted human serum were well mixed. Then, the mixture was added dropwise to the surface of PROD/PT-rGO-PtPd NCs/Au NPs/SPCE which was next placed in a 37 °C incubator for 40 min. After that, the working electrode was inserted into 5.0 mL of PBS solution. The level of 1,5-AG in the spiked samples was individually measured 3 times with DPV by using the developed 1,5-AG electrochemical biosensor.

## 3. Results and Discussion

### 3.1. Characterization of the PT-rGO-PtPd NCs

The morphology and composition of PT-rGO-PtPd NCs were obtained by TEM and SEM measurements. According to the TEM image (Figure 1A), some black spheres were observed to be attached to the membrane-like surface, indicating that Pt and Pd nanoparticles were adsorbed onto the PT-rGO. From the SEM image (Figure 1B), a film-like mesh yarn was found to be covered with many tiny white particles. Among these substances, the film-like mesh yarn is a PT-rGO composite, and the white particles with a size of about 100 nm are Pt or Pd NPs. The SEM image indicates that PT-rGO-PtPd NCs were successfully prepared and that the PtPd NPs were uniformly dispersed.

Figure 1C illustrates the UV-vis absorption spectroscopy of rGO (Figure 1C, curve a) and the PT-rGO-PtPd NCs (Figure 1C, curve b). rGO has two absorption peaks at 225 nm and 276 nm, while the PT-rGO-PtPd NCs only have one absorption peak at 227 nm. The fact that the peak (276 nm) was not observed in the PT-rGO-PtPd NCs solution suggests that the oxygen-containing functional group of rGO was reduced, and PtPd NPs were produced on the surface of rGO by the reducing agent in the solution. Moreover, the crystal structure of PT-rGO-PtPd NCs was analysed by XRD (Figure 1D). The diffraction peak at 2θ = 24.90° corresponds with the (002) crystal plane of rGO. The diffraction peaks at 2θ = 67.40°, 46.10°, and 39.70° represent (220), (200), and (111) of the Pt and Pd NPs, respectively. Pt and Pd NPs, which belong to the same family, were shown to have similar crystal structures as their peak levels were in proximity. All these peaks are in accordance with the JCPDS standard cards of Pt (PDF#04-0802)) and Pd (PDF# 46-1043), supporting that the Pt ions and Pd ions have been successfully formed Pt NPs and Pd NPs, and PT-rGO-PtPd NCs were well prepared.

### 3.2. Design Principle of the Electrochemical Determination of 1,5-AG Based on PT-rGO-PtPd NCs

PT-rGO-PtPd NCs are used not only as supporting materials which facilitate PROD immobilization in a stable and selective way through their large specific surface area and high density of active sites, but they also serve as electrochemical probes with an electrical signal and present a synergistic effect to electro-catalyze H_2_O_2_ oxidization. Figure 2A describes the analytical principle of the prepared biosensor for determining 1,5-AG. Firstly, PT-rGO-PtPd NCs were prepared by one-step reducing reaction via AA treatment. Then, SPCE was activated by H_2_SO_4_, followed by electrodeposited Au NPs. The as-prepared homogeneous PT-rGO-PtPd NCs were attached to the surface of Au NP/SPCE via π–π reaction and electrostatic adsorption. Next, a 1,5-AG electrochemical biosensor was constructed after PROD was immobilised on the surface of PT-rGO-PtPd NC/Au NP/SPCE. Lastly, 1,5-AG can be incubated in the sensing interface of the 1,5-AG electrochemical biosensor. The 1,5-AG/PROD/PT-rGO-PtPd NCs/Au NP/SPCE was used as the working electrode, and PBS was used as the supporting electrolyte. In the presence of 1,5-AG, PROD catalyzes the oxidization of 1,5-AG to generate 1,5-anhydrofuctose (1,5-AF) and hydrogen peroxide (H_2_O_2_). The generated H_2_O_2_ can be decomposed into H_2_O under the synergistic catalysis of PT-rGO-PtPd NCs which simultaneously enhance the redox reaction between PT and its oxidative product (quinones, PT_ox_). The redox peak current signal can be obtained and recorded by DPV method. Since redox peak current is directly proportional to the concentration of 1,5-AG, the level of 1,5-AG can be suggested by measuring the redox peak current with high sensitivity.

### 3.3. Electrochemical Performance of the Fabricated Electrodes

Figure 2B illustrates the DPV curves of different modified electrodes (PT-rGO-PtPd NCs/Au NPs/SPCE, 1,5-AG/PT-rGO-PtPd NCs/Au NPs/SPCE, PROD/PT-rGO-PtPd NCs/Au NPs/SPCE, and 1,5-AG/PROD/PT-rGO-PtPd NCs/Au NPs/SPCE) incubated in PBS solution. When the PT-rGO-PtPd NCs were modified onto the surface of Au NPs/SPCE (Figure 2B, curve a), a significant reduction peak at around −0.5 V was observed which could be explained by the electro-oxidization of PT. Moreover, the reduction peak in the current decreased after the addition of 1,5-AG (Figure 2B, curve b). This phenomenon could be caused by the enrichment of 1,5-AG, which is non-conductive, located on the electrode surface. Similarly, the reduction peak of the current also reduced after adding non-conductive PROD (Figure 2B, curve c). Furthermore, since PROD catalyzes the oxidization of 1,5-AG and further produces H_2_O_2_ which can be decomposed by PT-rGO-PtPd NCs, the reaction from PT to phenoxide on the surface of the electrode would be expectedly enhanced. Consequently, the redox peak current increased (Figure 2B, curve d).

Figure 2C shows a histogram of the electrochemical responses with different electrodes, where curve a, curve b, curve c, and curve d represent PT-rGO-PtPd NCs/Au NP/SPCE, 1,5-AG/PT-rGO-PtPd NCs/Au NP/SPCE, PROD/PT-rGO-PtPd NCs/Au NP/SPCE, and 1,5-AG/PROD/PT-rGO-PtPd NCs/Au NP/SPCE, respectively. An electrochemical signal can be obtained in the process of the electro-catalytic reaction between PT-rGO-PtPd NCs/Au NPs and H_2_O_2_. Our results suggested that 1,5-AG/PROD/PT-rGO-PtPd NC/Au NP/SPCE exhibited a higher current response (curve d, I = 26.73 μA) than the other electrodes (18.14 μA for curve a, 12.06 μA for curve b, and 17.36 μA for curve c) (Figure 2C). Since PROD stimulates 1,5-AG oxidization to generate H_2_O_2_ at the electrode interface and subsequently induces a current signal, a larger current signal represents more 1,5-AG captured by the sensor.

### 3.4. Electro-Catalytic Reaction between the PT-rGO-PtPd NCs and H_2_O_2_

To explore the electro-catalytic capacity of PT-rGO-PtPd NCs for H_2_O_2_, CV of the modified electrodes was performed at a scan rate of 50 mV/s in PBS solution in the presence of 100.0 μmol/L H_2_O_2_ as mediator. The characteristics of the CV spectra are oxidation and reduction peaks corresponding to the redox species present in the working electrode.

As shown in Figure 3A, the SPCE exhibited no identifiable redox current response (Figure 3A, curve a), implying that there was no electro-catalytic reduction of H_2_O_2_ in the SPCE. For the Au NPs/SPCE (Figure 3A, curve b), a pair of redox peaks were observed, suggesting that Au NPs induce the electro-catalytic reduction of H_2_O_2_ in a neutral solution. Moreover, a mild ladder-like cathodic response which presents H_2_O_2_ reduction was noticed. In the control group, the PT-rGO-PtPd NCs/Au NPs/SPCE (Figure 3A, curve c) had a distinct reduction signal of H_2_O_2_ at −0.5 V, which indicates that the proposed PT-rGO-PtPd NCs/Au NPs/SPCE significantly enhanced the electro-catalytic reduction of H_2_O_2_. Based on the results illustrated above, it could be inferred that PT-rGO-PtPd NCs show good performance in electro-catalytically reducing H_2_O_2_ and subsequently detecting 1,5-AG levels.

### 3.5. Electrochemical Characterization of Different Electrodes

The EIS technique was used to characterise the stepwise fabrication process of the modified electrodes. Figure 3B presents the Nyquist diagrams and Randall equivalent circuit of different modified electrodes. The circuit involves electron transfer resistance (Ret), the resistance of the solution (Rs), Warburg impedance (Zw), and double-layer capacitance (Cp). The Ret value is estimated from the diameter of the semicircular portion at high frequencies in the Nyquist plot. In Figure 3B, each diagram has a semi-circle that corresponds to the charge transfer resistance (Ret) of the [Fe (CN)_6_]^4−^/[Fe (CN)_6_]^3−^ reaction. Specifically, the Ret value of the SPCE electrode (Figure 3B, curve a) was measured to be 1849 Ω, while the other modified electrodes showed lower Ret values. The Ret of Au NPs/SPCE was 85 Ω (Figure 3B, curve b) because of electron transfer facilitated by Au NPs. Moreover, the Ret of PT-rGO-PtPd NC/Au NP/SPCE was 63 Ω (Figure 3B, curve c) due to the good electron transfer capacity shared with both PT-rGO-PtPd NCs and Au NPs. The Ret substantially increased (Figure 3B, curve d, 257 Ω) after PRODs were attached to the surface of the PT-rGO-PtPd NC sheets. Furthermore, the diameter of the semicircle increased (Figure 3B, curve e, 754 Ω) when 1,5-AG was added to the electrode surface. The interaction between the target/enzyme complex and the active sites on the electrode surface would increase resistance and subsequently the Nyquist diameter.

The electrical conductivity and the effective surface area of the modified electrodes constructed were also investigated by CV in PBS solution (0.1 mol/L, pH 7.0) containing 5.0 mmol/L K_3_Fe (CN)_6_/K_4_Fe (CN)_6_ and 0.1 mol/L KCl solution. Firstly, as shown in Figure 3C, a pair of redox peaks with an anodic peak current (Ipa) of 48.44 μA and a peak potential difference (ΔEp) of 0.154 V were observed at the surface of the SPCE (Figure 3C, curve a). Secondly, for Au NPs/SPCE, the signal was extensively intensified (Ipa = 83.15 μA) and ΔEp increased (ΔEp = 0.172 V) (Figure 3C, curve b), indicating that Au NPs promote the electrochemical redox of the sensor. Thirdly, after the PT-rGO-PtPd NCs were modified to the surface of Au NPs/SPCE, the anodic peak current decreased slightly (Ipa = 79.95 μA) and ΔEp decreased (ΔEp = 0.154 V), while the anodic peak current substantially increased (Ipa = 89.55 μA) (Figure 3C, curve c). It can be speculated that PT-rGO-PtPd NCs have good conductivity and a large specific surface area to provide more reaction sites for [Fe (CN)_6_]^3−^/^4−^. Fourthly, after the PRODs were covalently immobilised on the PT-rGO-PtPd NC/Au NP/SPCE, a decreased redox peak current and an increased ΔEp were observed (Ipa = 69.33 μA; ΔEp = 0.220 V) (Figure 3C, curve d), which can be explained by the fact that PROD is a type of protein that hinders electron transfer. Lastly, for 1,5-AG/PROD/PT-rGO-PtPd NCs/Au NPs/SPCE, both the peak current and ΔEp decreased (Ipa = 62.25 μA; ΔEp = 0.168 V) (Figure 3C, curve e), indicating that PROD successfully interacts with 1,5-AG. These results indicated that the sensing interface was well synthesised.

Moreover, the effective surface area of all of the modified electrodes was determined by utilising the CV result of [Fe (CN)_6_]^3−/4−^, derived from Randles-Sevcik’s formula as the following Formula (1).
(1)$$I_{P}=2.69\times 10^{5}AD^{1/2}n^{3/2}v^{1/2}C$$

In this formula, *I_P_* represents the current peak (μA), *A* represents the effective surface area of modified electrodes (cm^2^), *D* is the medium proliferation parameter [Fe (CN)_6_]^3−/4−^ (6.70 × 10^−6^ cm^2^ s^−1^), *n* is the number of electrons involved ([Fe(CN)_6_]^3−/4−^, *n* = 1), *v* represents the scanning rate (0.1 V/s), and *C* represents the concentration of redox medium (5.0 mmol/L).

The effective surface area of different electrodes was calculated with Randles-Sevcik’s formula and placed in the following order: SPCE (0.0440 cm^2^) < 1,5-AG/PROD/ PT-RGO-Pt-Pd NCs/Au NPs/SPCE (0.0565 cm^2^) < PROD/PT-RGO-Pt-Pd NCs/Au NPs/SPCE (0.0630 cm^2^) < PT-RGO-Pt-Pd NCs/Au NPs/SPCE (0.0726 cm^2^) < Au NPs/SPCE (0.0755 cm^2^). This result verified that PT-RGO-Pt-Pd NCs and Au NPs significantly increase the effective surface area of the electrode which facilitates electron transfer in the electrochemical reaction.

### 3.6. Raman Characterization of the 1,5-AG Biosensor

Raman spectroscopy is a commonly-used characterization method to detect the change of carbon elements in two main characteristic peaks, D and G. It can also be used to analyse the stepwise fabrication process of modified electrodes. Figure 3D illustrates the Raman spectra of different stages of SPCE modification: (a) SPCE, (b) Au NPs/SPCE, and (c) PT-rGO-PtPd NCs/Au NPs/SPCE. Compared with a bare SPCE electrode (Figure 3D, curve a), the ratio of its D-band to G-band became smaller when the SPCE had been deposited with Au NPs (Figure 3D, curve b), indicating that the electrode was successfully modified with Au NPs. Moreover, when the electrodes were further modified with PT-rGO-PtPd NCs, it can be observed that the intensity of the D-band and G-band of PT-rGO-PtPd NCs/Au NPs/SPCE (Figure 3D, curve c) (I_D_/I_G_ = 0.942) was the highest compared with the other stages. These results indicated that the PT-rGO-PtPd NCs/Au NPs/SPCE was well constructed.

### 3.7. Optimization of the Experimental Conditions

The DPV technique was chosen for 1,5-AG detection in this study. The DPV technique is applied with linear sweep voltammetry with a series of regular voltage pulses superimposed on the linear potential sweep. The current was measured immediately before each potential change so that the effect of charging current could be minimised, and higher sensitivity can be achieved. Moreover, a short time (about 30 s) is enough for DPV to complete its measurement, while 600 s or more is required for other techniques such as EIS and AC impedance measurement [38,39]. To achieve the optimal analytical performance of the 1,5-AG biosensor, some experimental parameters, including incubation time, incubation temperature, the amount of PT-rGO-PtPd NCs, and the concentration of PROD, were explored.

Firstly, the optimal incubation time was investigated. The DPV response continuously and markedly increased before it reached its peak after 40–60 min of incubation (Figure 4A). It could be inferred that the H_2_O_2_ decomposition, which is catalyzed by the PROD enzyme and 1,5-AG, reached its maximum after 40 min of incubation so no greater DPV signal could be observed.

Next, we assessed the optimal incubation temperature. Since the optimal temperature for human enzymes to perform their enzymatical function is around 37 °C, the PROD enzyme is expected to perform its strongest catabolic activity under a 37 °C temperature to produce H_2_O_2_. In Figure 4B, the electrochemical signal sharply increased as the temperature increased from 0 to 37 °C, and it substantially dropped when the temperature continued to increase as expected. Consequently, the optimal incubation temperature was set as 37 °C to exert the highest DPV response in the following experiments.

Moreover, we sought to identify the appropriate amount of PT-rGO-PtPd NCs to obtain the strongest electrochemical signal. The response current of the 1,5-AG biosensor rose when the amount of PT-rGO-PtPd NCs increased from 3.0 to 9.0 μL; however, it slightly decreased as the amount of PT-rGO-PtPd NCs grew from 9.0 to 18.0 μL (Figure 4C). It can be safely concluded that 9.0 μL would be the appropriate amount of PT-rGO-PtPd NCs to ensure good performance by the 1,5-AG biosensor.

Lastly, the best concentration of PROD for the 1,5-AG biosensor was explored. Initially, the electrochemical signal became stronger synchronously with the increasing concentrations of PROD, and it reached its maximum value when 0.5 mg/mL of PROD was added, indicating that 0.5 mg/mL is the appropriate concentration for PROD to perform its catalytic activity. Later, the electrochemical signal decreased when the concentration of PROD continued to increase from 0.5 to 2.0 mg/mL (Figure 4D). Since PROD is non-conductive, the resistance to electron transfer would gradually increase if more PRODs continued gathering, resulting in a lower DPV response.

In summary, to achieve a satisfied electrochemical signal for the 1,5-AG biosensor, 40 min, 37 °C, 9.0 μL, and 0.5 mg/mL were chosen for the incubation time, the incubation temperature, the amount of PT-rGO-PtPd NCs, and the concentration of PROD, respectively. These parameters were used as the optimal criteria in the following experiments.

### 3.8. Analytical Performance of the 1,5-AG Biosensor

Under the experimental conditions detailed above, DPV measurement was used to evaluate the analytical performance of the 1,5-AG biosensor with different concentrations of 1,5-AG. As shown in Figure 5A, the peak current of DPV response increased along with increasing concentrations of 1,5-AG. This phenomenon can be explained by the fact that more H_2_O_2_ would be produced when PRODs in the sensitive interface react with more 1,5-AG, leading to a higher electron transfer rate in the decomposition process, a simultaneously stronger redox reaction between PT and its oxidative product (quinones, PT_ox_), and a subsequently higher current signal. Moreover, when the concentration of 1,5-AG was in the range of 0.1 to 2.0 mg/mL, a linear relationship between the peak current (µA) and the concentration of 1,5-AG was found on the current-potential curve of DPV (Figure 5B). The linear regression equation of the fabricated aptasensor is I (μA) = 6.83 C_1,5-AG_ + 39.99 (R^2^ = 0.9996). The limit of detection (LOD) value, defined by the equation 3Sb/S (Sb: the standard deviation of the blank (*n* = 5), S: the slope of the calibration curve), was calculated to be 30.0 μg/mL.

Moreover, we further compared our proposed 1,5-AG biosensor based on PT-rGO-PtPd NCs with other reported 1,5-AG detection methods in Table 1. In terms of detection range, our prepared 1,5-AG biosensor possesses a wider linear range compared with other reported methods. Although our biosensor has many advantages, including a simple structure, being easily synthesised, and being manufactured in bulk at a lower cost, our 1,5-AG biosensor has a higher LOD that may be caused by the limited affinity between 1,5-AG and PROD and slower electron transfer from the electrode due to the poor conductivity of 1,5-AG and PROD, leading to a low amount of variation in the DPV signal. In the detection of actual 1,5-AG samples, a higher LOD is inadequate as the 1,5-AG concentrations are 12–40 μg/mL in normal human blood and decrease in DM patients. Based on the current study, to overcome these shortcomings, aptamers and other nanometer materials with good electrochemical activity, instead of PROD, will be considered to recognise 1,5-AG and to reduce the LOD.

### 3.9. Selectivity, Stability, and Reproducibility of the 1,5-AG Biosensor

The selectivity of electrochemical aptasensors is crucial in bio-analysis. To evaluate the selectivity of our developed biosensor, we tested the sensing response of our biosensor in PBS solution containing some interfering biological molecules, such as glucose, ascorbic acid, uric acid, and cholesterol, which are present in human blood. The concentration of these biological molecules was 10.0 mg/mL, which is ten times the concentration of 1,5-AG (1.0 mg/mL). The maximum percentage of biological molecules in the PBS solution was less than 28.1% which is under the acceptable limit. As shown in Figure 5C, the sensing responses were 7.7 μA for glucose, 13.3 μA for ascorbic acid, 8.9 μA for uric acid, 10.9 μA for cholesterol, and 47.4 μA for 1,5-AG. These results indicated that our biosensor exhibits a higher affinity to 1,5-AG even though the concentration of other interfering biological molecules is much higher than that of 1,5-AG.

Moreover, the repeatability of our biosensor was verified by five replicated experiments to detect 0.3 mg/mL of 1,5-AG; the RSD was 2.68%. The reproducibility of our proposed method was also investigated. Five independent measurements with five separate electrodes, which were prepared under the same fabrication conditions, were used to detect 0.3 mg/mL of 1,5-AG. The corresponding RSD was calculated as 5.65%. These results demonstrated that our proposed biosensor has acceptable reproducibility.

Furthermore, our proposed biosensors were stored in a fridge at 4 °C for 3 days and 7 days before being used to test the samples with 0.3 mg/mL of 1,5-AG. The corresponding DPV response currents were recorded to investigate the stability of our 1,5-AG biosensor. After 3 days and 7 days of storage, the values of the current response of experimental biosensors were 97.5% and 93.3% of the initial value of the control biosensors, respectively. The results indicated that our proposed biosensor also has acceptable stability.

### 3.10. Application of Our 1,5-AG Electrochemical Biosensor in Human Serum Samples

To evaluate the practicability of our 1,5-AG biosensor in human serum samples, recovery experiments were first carried out by adding different concentrations of 1,5-AG to a kind of normal human serum sample. The DPV technique was then applied to determine the 1,5-AG level by using our biosensor. The results are summarised in Table 2.

## 4. Conclusions

In this study, an electrochemical biosensor to determine 1,5-AG was well synthesised by the covalent attachment of PROD to PT-rGO-PtPd NCs which acts as a transduction platform. Coordinately, PT-rGO-PtPd NCs are used as mimetic enzymes and exhibit a strong catalytic effect on H_2_O_2_. In the presence of 1,5-AG, PROD catalyzes the oxidization of 1,5-AG to generate H_2_O_2_ which can be decomposed into H_2_O under the synergistic catalysis of PT-rGO-PtPd NCs. Then, the redox reaction between PT and its oxidative product (quinones, PTox) can be enhanced simultaneously by PT-rGO-PtPd NCs, and the current signal is recorded by the DPV method. Moreover, the developed biosensor has a linear dynamic correlation with 1,5-AG in the range of 0.1–2.0 mg/mL, with an LOD of 30.0 μg/mL. Furthermore, we demonstrated that our developed biosensor exhibits high selectivity, acceptable stability, and good recovery in human serum samples. However, the limitation in this study is that our biosensor has a higher LOD compared with other reported biosensors. To obtain the satisfied LOD, other nanometer materials with good catalytic and electrochemical activity would be considered to modify the electrochemical biosensor, and aptamer would be explored to recognize 1,5-AG. All in all, this fabrication strategy sheds light on a promising application that could be modified as a portable device for fast and precise detection of 1,5-AG.

## Figures and Tables

**Figure 1 materials-16-02786-f001:**
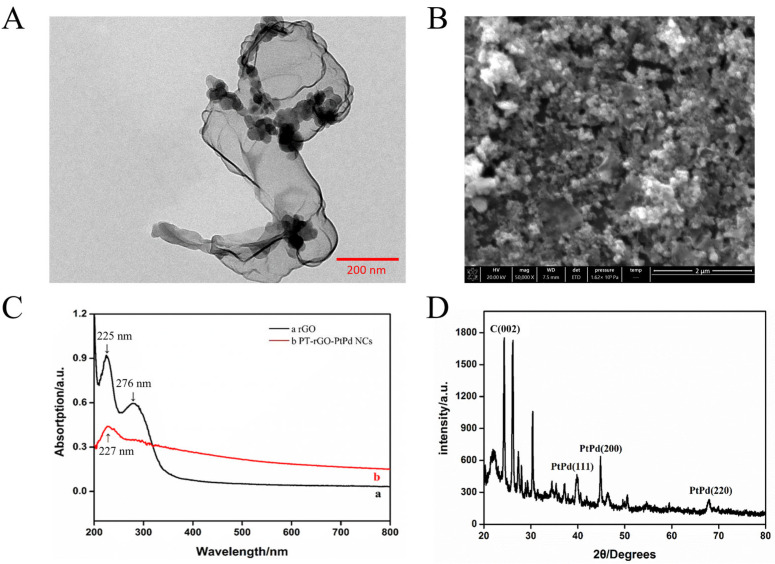
(**A**) The TEM image of PT-rGO-PtPd NCs. (**B**) The SEM image of PT-rGO-PtPd NCs. (**C**) The UV-visible spectrum of the rGO (a) and PT-rGO-PtPd NCs (b). (**D**) The XRD spectra of PT-rGO-PtPd NCs.

**Figure 2 materials-16-02786-f002:**
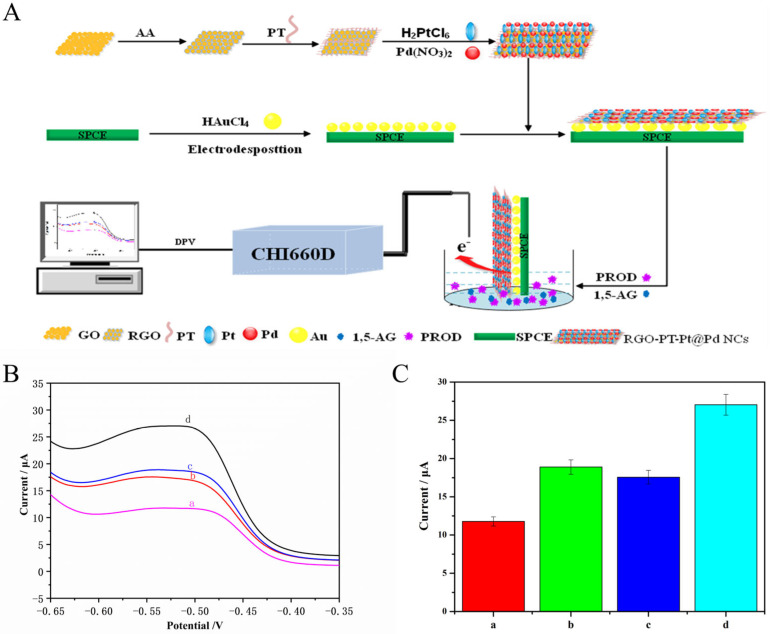
(**A**) Schematic diagram of an electrochemical sensor based on PT-rGO-PtPd NCs for 1,5-AG detection. (**B**) DPV maps of the feasibility of 1,5-AG detection performed by different electrochemical sensors in a PBS solution (0.1 mol/L; pH 7.0). (a) PT-rGO-PtPd NCs/Au NPs/SPCE, (b) 1,5-AG (0.1 mg/mL) /PT-rGO-PtPd NCs/Au NPs/SPCE, (c) PROD/PT-rGO-PtPd NCs/Au NPs/SPCE, (d) 1,5-AG (0.1 mg/mL)/PROD/PT-rGO-PtPd NCs/Au NPs/SPCE. (**C**) Histogram of the feasibility of electrochemical sensor for determining 1,5-AG. ((a) PT-rGO-PtPd NCs/Au NPs/SPCE, (b) 1,5-AG (0.1 mg/mL)/PT-rGO-PtPd NCs/Au NPs/SPCE, (c) PROD/PT-rGO-PtPd NCs/Au NPs/SPCE, (d) 1,5-AG (0.1 mg/mL)/PROD/PT-rGO-PtPd NCs/Au NPs/SPCE).

**Figure 3 materials-16-02786-f003:**
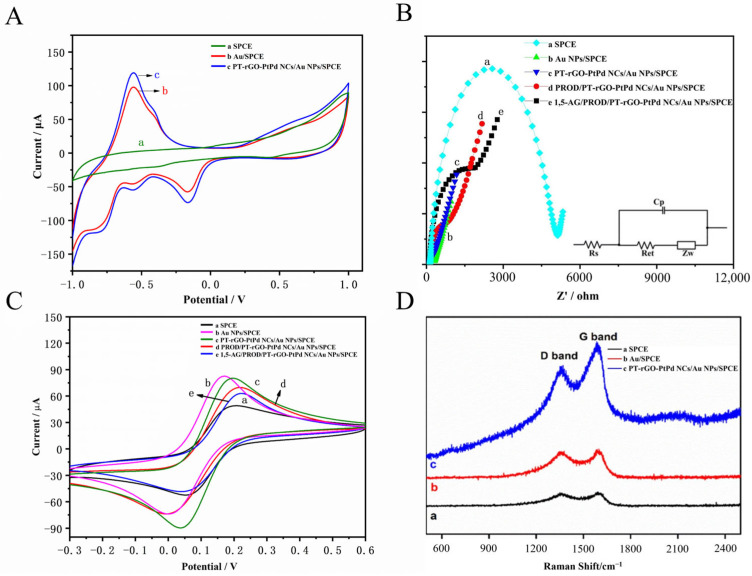
(**A**) The CV curves of (a) SPCE, (b) Au NPs/SPCE, and (c) PT-rGO-PtPd NCs/Au NPs/SPCE in PBS solution (0.1 mol/L, pH 7.0) in the presence of 100.0 μmol/L H_2_O_2_. (**B**) EIS and (**C**) CV curves of different electrodes ((a) SPCE, (b) Au NPs/SPCE, (c) PT-rGO-PtPd NCs/Au NPs/SPCE, (d) PROD/PT-rGO-PtPd NCs/Au NPs/SPCE, and (e) 1,5-AG/PROD/PT-rGO-PtPd NCs/Au NPs/SPCE) in PBS solution (0.1 mol/L, pH 7.0) containing 5.0 mmol/L K_3_Fe (CN)_6_/K_4_Fe (CN)_6_ and 0.1 mol/L KCl. (**D**) The Raman spectrum of different modified electrodes ((a) bare SPCE, (b) Au NPs/SPCE, and (c) PT-rGO-PtPd NCs/Au NP/SPCE).

**Figure 4 materials-16-02786-f004:**
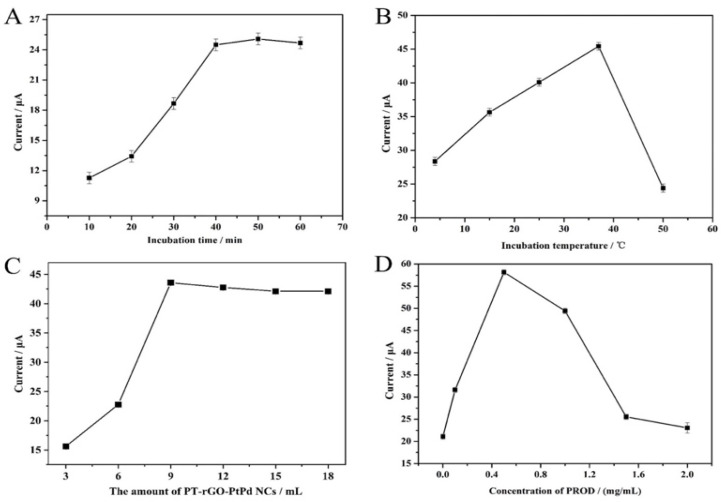
The optimization of experimental conditions. The effects of (**A**) different incubation times, (**B**) different incubation temperatures, (**C**) different amounts of PT-rGO-PtPd NCs, and (**D**) different concentrations of PROD on the response of the 1,5-AG biosensor were explored by the DPV method in PBS solution (0.1 mol/L, pH 7.0). The concentration of 1,5-AG in this experiment was set to 1.0 mg/mL. The error bars represent relative standard deviations (RSDs) (*n* = 3).

**Figure 5 materials-16-02786-f005:**
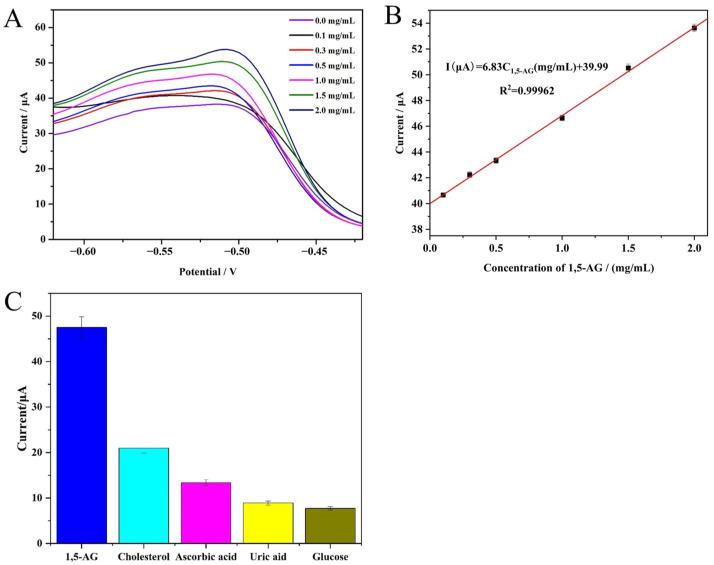
(**A**) The DPV plots of 1,5-AG at increasing concentrations from 0.1 to 2.0 mg/mL were obtained by using the 1,5-AG electrochemical biosensor. (**B**) The linear calibration curve of the electrochemical biosensor in the range of 0.1 to 2.0 mg/mL of 1,5-AG. (**C**) The histogram of the specificity of the proposed 1,5-AG electrochemical biosensor. The error bars represent the RSD (*n* = 3).

**Table 1 materials-16-02786-t001:** A comparison between our proposed 1,5-AG electrochemical biosensor and other reported biosensors regarding sensing characteristics.

Sensing Method	Linear Range	Detection Limit	Reference
Enzymatic method	0–100 mg/L	0.3 mg/L	[9]
^a^ EIS	0.8–12 mg/dL; 10–300 mg/dL	0.8 mg/dL	[10]
Glycomark™ assay	4.0–16.0 μg/mL	-	[11]
^b^ LC-MS/MS	1–50 μg/mL	0.2 μg/mL	[12]
Colourimetric assay	20.0–100.0 μg/mL	0.144 μg/mL	[13]
^c^ LAPS biosensor based on enzymatic silver deposition	60–225 μg/mL	40 μg/mL	[40]
^c^ LAPS biosensor on rGO-PAMFc/AuNPs sensing membrane	100–1000 μg/mL	21.74 μg/mL	[41]
^d^ HPLC-BA	100 ng/mL–50 μg/mL	10 ng/mL	[42]
^e^ LC/MS^3^	50 ng/mL–10 μg/mL	-	[43]
^f^ OFET-based biosensor	0–10 mM	1 mM (164 μg/mL)	[44]
Paper-based colourimetric assay	0–50 μg/mL	3.2 μg/mL	[45]
PT-rGO-PtPd NCs-based electrochemical biosensor	100–2000 μg/mL	30.0 μg/mL	This work

Abbreviation: ^a^ EIS, electrochemical impedance spectroscopy; ^b^ LC-MS/MS, liquid chromatography–tandem mass spectrometry; ^c^ LAPS, light-addressable potentiometric sensor; ^d^ HPLC-BA, high-performance liquid chromatography with benzoic acid derivatization; ^e^ LC/MS^3^, liquid chromatography/triple quadrupole mass spectrometers; ^f^ OFET-based biosensor, organic field-effect transistor-based biosensor.

**Table 2 materials-16-02786-t002:** The detection of 1,5-AG in human serum samples with our proposed biosensor.

Sample Type	1,5-AG Concentration added (mg/mL)	1,5-AG Concentration Identified (mg/mL)	Recovery (%)	RSD (*n* = 5, %)
Normal human serum	0.25	0.267	106.80	1.65
	0.5	0.519	103.80	7.32
	1.0	0.998	99.80	4.40

## Data Availability

The data are available upon reasonable request from the corresponding author.

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
