# Peer review of "An Efficient Electrochemical Biosensor to Determine 1,5-Anhydroglucitol with Persimmon-Tannin-Reduced Graphene Oxide-PtPd Nanocomposites"

_materials, 2023, doi:10.3390/ma16072786_

Round 1

Reviewer 1 Report (Previous Reviewer 3)

Please explain to me how useful a biosensor is whose detection limit for a certain analyte is 0.030 mg/mL, and that molecule is present in values of 0.012-0.040 mg/mL in healthy patients and even lower in sick patients, as you yourselves state in lines 43-44. This biosensor has no practical application, so this research is neither novel nor useful.

Author Response

Reviewer 2 Report (Previous Reviewer 4)

The paper has been mainly revised with necessary corrections and amendments according to all the reviewers' comments. The resulting manuscript is more satisfying.

Author Response

Reviewer 3 Report (New Reviewer)

The authors described an approach for an Efficient Electrochemical Biosensor to Determine 1,5-Anhy-droglucitol with Persimmon Tannin–Reduced Graphene Ox ide–PtPd Nanocomposites” is an interesting work and nicely presented. The materials characterization procedures and data are presented well. I recommend the paper for publication after the authors made the following Minor corrections/additions.

I advise the authors to take the following points into account while revising their manuscript.

Comment 1: There are some typographical and grammatical errors, in the manuscript text, so the authors need to correct them in the revised manuscript.

Comment 2: English needs to be a little improved, as there are some misused conjunctions and technical flaws. So it needs to be corrected in the manuscript.

Comment 3: The Abstract needs to be revised, not so informative, let the author elaborate and explain the research question clearly, also mention the performed characterization techniques in the Abstract section.

Comment 4: The authors procured Graphene oxide from XFNANO Materials Technology, and they did not mention in the manuscript how they prepared graphene oxide to Reduced graphene oxide. So the authors need to add the reduced graphene oxide procedure in the revised manuscript.

Comment 5: The authors performed SEM and TEM analysis of the nanocomposite. So I would suggest authors include EDX analysis to assess the elemental composition of the nanocomposite.

Comment 6: Please improve the resolution of all figures. Also, enhance the  schematic diagram of an electrochemical sensor based on PT–rGO–PtPd NCs (Figure 2A)

Comment 7: The homogeneity of the reference section needs to be maintained. So please check and revise accordingly to the journal's instructions.

Round 2

Reviewer 1 Report (Previous Reviewer 3)

It could be published in the present form

This manuscript is a resubmission of an earlier submission. The following is a list of the peer review reports and author responses from that submission.

Round 1

Reviewer 1 Report

Conceptually, this is an interesting paper in which the authors use their reported non-enzymatic electrochemical hydrogen peroxide biosensor for the enzymatic determination of 1,5-2 Anhydroglucitol. Figure 5 seems to be the key result, but I am having difficultly interpreting the result in light of the previous report in ref 37. Why is DPV needed and not amperometry? Is the peak in Figure 5A an oxidation or reduction? It looks as if an electrochemical sign convention has been switch compared to ref 37. 

Reviewer 2 Report

materials-2078490

The manuscript authored by Li et al. entitled "An Efficient Electrochemical Biosensor of Determination of 1,5-Anhydroglucitol with Persimmon Tannin-Reduced Graphene Oxide-PtPd Nanocomposites". The goal of this study is to apply electrochemical biosensor based on persimmon tannin-reduced graphene oxide-Pt-Pd nanocomposites (PT-rGO-PtPd NCs) to specifically detect the level of 1,5-AG. The study is original research, described in sufficient detail and add the conclusions a reasonable extension of the results. These considerations, in conjunction with the wide interest of the biosensors research, make the manuscript of interest for the broad readership of Materials. I recommend this paper for publication after minor revision acceding to the following points:

1.      In the Abstract section (line 20), the authors should rewrite H2O2 using standard typography for chemical compounds.

2.      In the Introduction section (line 42), the authors mention that the sensor has a sensitivity limit of detection of 0.03 mg/mL. However, this does not match with the normal concentration of 1,5-AG, which is approximately 12-40 μg/mL (as stated in the Abstract section, line 42). This means that it is not possible to measure 1,5-AG in blood serum using this sensor. Additionally, the authors have provided a incorrect value in Table 1 (0.03 μg/mL).

3.      The authors mention that the synthesized rGO-PT-Pt has a low detection limit of 0.26 μM (Introduction section, line 77). How did the authors calculate this value in micromolar units?

4.      The authors should include legends for Figure 1 (C) to distinguish between the black and red lines. The explanation for these lines is provided in the Results and Discussion section, line 177.

5.      In Figure 2 (B) and (C), the authors should include legends for the different electrodes to distinguish between the a, b, c, and d lines. They should also provide names for each electrode. The explanation for these electrodes is provided in the Results and Discussion section, line 221. Similarly, the authors should also include legends with the names of the four electrodes in Figure 3.

Reviewer 3 Report

I believe that this article does not deserve to be published in Materials journal since it is a highly researched topic and it does not improve the scientific knowledge generated by the corresponding investigations.

The authors affirm (line 357) that this investigation provides better LOD than other works, but it is not true (there is a erratum in the Table 1, their LOD = 0.03 mg/mL not 0.03 ug/mL).

The figures have to be greatly improved for the article to be accepted in a scientific journal.

Reviewer 4 Report

In this study, an electrochemical biosensor, based on persimmon tannin-reduced graphene oxide-Pt-Pd nanocomposites (PT-rGO-PtPd NCs) onto the surface of the screen-printed electrode (SPCE), has been designed for the 1,5-Anhydroglucitol (1,5-AG) detection. The materials obtained were characterized by UV-Vis spectroscopy, Raman spectroscopy, SEM and TEM.

Overall, the experimental work appears to be carried out according to scientific standards. However, after careful examination, I can not recommend its publication in Materials. The authors should fully take the following points into account and revise their manuscript prior to further submission.

1.         It is strongly suggested to check the grammatical mistakes in the manuscript. For example line 209 “Sine” maybe “since”? In the Abstract use subscripts for hydrogen peroxide, Id/Ig should be ID/IG, etc.

2.         In the Introduction part, the Authors represent their results (Fig. 1). This figure should be in the results and discussion section. In the Introduction part the author should present only the novelty, problem and aim of the research.

3.         The Authors need to present the purity of the reagents used in this study.

4.         The Authors need to present the parameters of measurement (voltage, magnification, etc. ) as well as the sample preparation procedure for the measurements.

5.         The Authors need to present more information about commercial SPCE electrodes (e.g. dimensions)

6.         Why did the Authors change the pH of PBS from 7 to 7.4 during experiments?

7.         Why the deposition of AuNPs onto the SPCE was performed? Why SPCE was not directly coated with PT-rGO-PtPd NCs? Maybe for the electrochemical response responsible are AuNPs not your prepared composite?

8.         Non-explicitness should be avoided. Kindly specify this "different concentration" in the 2.5 section, “with different electrodes” in line 218.

9.         The discussion about structural and electrochemical results is poor. There is a lack of depth analysis of the reasons behind the sensors' good performance in several parts of the manuscript. The authors do not compare samples with each other. The authors do not provide any explanation why one is better than others. The authors should add a more detailed discussion regarding the structure-performance relationship of the materials.

10        The authors write about the specific surface area, however, do not provide the values. What is the electroactive surface area for all electrodes?

11        the interpretation of the structural data (e.g. Raman, XRD) is very superficial, incomplete and incomprehensible.

12. I recommend the authors compare their results with the results of other research groups.

13.       The Authors should provide XRD PDF standard cards which were used or references.

14.       D-band and the G-band of the PT-rGO-PtPd NCs/SPCE (curve c) (Id/Ig=0.942) were found to be the highest intensity compare with other stages. So what?

15.       All the redox peaks in CVs in Fig. 3 A should be assigned and mentioned.

16.       What is the physical meaning of 106.8%?

17.       The caption of fig. 2 should be similar to Fig.3 Kindly, specify the concentration of 1,5-AG and a,b,c,d letters

18. Lines 196-206 (Results and discussion) repeat the synthesis and the electrode preparation procedures, which previously were described in the 2.3, 2.4, and 2.5 sections. Kindly avoid this. In the results section of your academic paper, you should present what you found when you conducted your analyses, whereas in your discussion section you explain what your results mean and connect them to prior research studies.

19.       All curves in DPV (independent of the sample) have a similar shape, and no peak a visible, only the background curve was increased. How the Authors can make the conclusions that their proposed materials are sensitive to the 1,5-AG detection?

20.       EIS data are commonly analyzed by fitting to an equivalent electrical circuit model. Kindly provide the electrical circuit model.

Round 2

Reviewer 1 Report

The authors have answered my questions.

Reviewer 3 Report

My decision has not changed.

Only one of the reviewed papers contributed by the authors has a higher LOD than the one reported in this paper, therefore it is NOT more sensitive than those papers to which the authors refer.

Reviewer 4 Report

In the revised version, the authors respond to the questions raised by the reviewers, modified some descriptions and provide supplementary material. However, the discussions of the electrochemical section still remain some major problems without resolved, for example, verifying the effect of dissolved oxygen, and assigning the broad wave in DPV (Fig. 5A).

Why did the authors obtain the conclusion that the smooth “peak” at -0.50V is for 1,5-AG, without making clear what the redox peak is? Is it originated from functional groups on modified SPCE with PT–rGO–PtPd NC or the reduction/oxidation of 1,5-AG? The Violet curve (0,0 mg/mL of 1,5-AG) has the same broad wave in the DPV. After the addition higher concentration of 1,5-AG, no peak a visible, only the background curve was increased. It seems, that this peak is originated from functional groups on modified SPCE with PT–rGO–PtPd NC.

In the revised article, I still do not find the information about the SPCE surface area or dimensions.

Randles-Sevcik's formula should be (3) not (1).

In view of those questions, I do not recommend it for publication in its present form.
